

# Permutation-based methods for mediation analysis in studies with small sample sizes

Miranda E. Kroehl[1], Sharon Lutz[2] and Brandie D. Wagner[1]

[1] Department of Biostatistics and Informatics, Colorado School of Public Health, University of Colorado Anschutz Medical Campus, Aurora, CO, USA
[2] Department of Population Medicine, Harvard Medical School and Harvard Pilgrim Health Care, Harvard University, Boston, MA, USA

## ABSTRACT

**Background:** Mediation analysis can be used to evaluate the effect of an exposure on an outcome acting through an intermediate variable or mediator. For studies with small sample sizes, permutation testing may be useful in evaluating the indirect effect (i.e., the effect of exposure on the outcome through the mediator) while maintaining the appropriate type I error rate. For mediation analysis in studies with small sample sizes, existing permutation testing methods permute the residuals under the full or alternative model, but have not been evaluated under situations where covariates are included. In this article, we consider and evaluate two additional permutation approaches for testing the indirect effect in mediation analysis based on permutating the residuals under the reduced or null model which allows for the inclusion of covariates.

**Methods:** Simulation studies were used to empirically evaluate the behavior of these two additional approaches: (1) the permutation test of the Indirect Effect under Reduced Models (IERM) and (2) the Permutation Supremum test under Reduced Models (PSRM). The performance of these methods was compared to the standard permutation approach for mediation analysis, the permutation test of the Indirect Effect under Full Models (IEFM). We evaluated the type 1 error rates and power of these methods in the presence of covariates since mediation analysis assumes no unmeasured confounders of the exposure–mediator–outcome relationships.

**Results:** The proposed PSRM approach maintained type I error rates below nominal levels under all conditions, while the proposed IERM approach exhibited grossly inflated type I rates in many conditions and the standard IEFM exhibited inflated type I error rates under a small number of conditions. Power did not differ substantially between the proposed PSRM approach and the standard IEFM approach.

**Conclusions:** The proposed PSRM approach is recommended over the existing IEFM approach for mediation analysis in studies with small sample sizes.

Corresponding author
Miranda E. Kroehl,
miranda.kroehl@cuanschutz.edu

## INTRODUCTION

Mediation analysis can be used to evaluate whether the exposure acts on the outcome through an intermediate variable (i.e., the mediator). The product of two regression

coefficients can be used to estimate and test this indirect effect (*Baron & Kenny, 1986*). However, because the product of two normally distributed random variables, such as two regression coefficients, is not normally distributed and does not approximate commonly used distributions, testing the indirect effect requires careful consideration (*Aroian, 1944*).

There have been a variety of tests proposed to evaluate the indirect effect in mediation analysis, including parametric methods (*MacKinnon et al., 2002*; *MacKinnon, Lockwood & Williams, 2004*; *Williams & MacKinnon, 2008*; *Hayes, 2009*; *Taylor & MacKinnon, 2012*; *Koopman et al., 2015*) and resampling methods (*VanderWeele, 2014*; *MacKinnon et al., 2002*; *MacKinnon, Lockwood & Williams, 2004*; *Williams & MacKinnon, 2008*; *Hayes, 2009*; *Taylor & MacKinnon, 2012*; *Koopman et al., 2015*). However, for studies with small sample sizes, bootstrapping approaches can result in inflated type I error rates and permutation testing has been proposed as an alternative resampling approach (*Koopman et al., 2015*; *Tofighi & MacKinnon, 2011*; *Williams & MacKinnon, 2008*).

Special consideration is needed for permutation testing in mediation analysis. Due to the assumption of no unmeasured confounding in mediation analysis, covariates need to be included to account for any confounding of the exposure-mediator-outcome relationships. The inclusion of covariates can add complexity to permutation approaches and the standard approach of permuting raw values is not appropriate for this situation. Permuting the outcome not only breaks the association between the exposure and outcome, but additionally breaks up the association between the outcome and any confounders, yielding a global test of all regression coefficients rather than the coefficient of interest (*Anderson & Legendre, 1999*; *Freedman & Lane, 1983*). Similarly, permuting the exposure will not only break the association between the exposure and outcome, but would also break the associations between the exposure and any confounders. An alternative to permuting the outcome or the exposure is to use the residuals from a linear model as the permutable units for the test (*Anderson & Legendre, 1999*). Two major approaches in the field are to permute the residuals under the reduced, or null, model (*Freedman & Lane, 1983*) which excludes the covariate of interest, or permute the residuals under the full, or alternative, model which includes the covariate of interest (*Manly, 1997*; *ter Braak, 1992*). For example, consider a test on the regression coefficient $\beta_1$, where $\beta_1$ measures the linear association between covariate $C_1$ and outcome $Y$ while accounting for covariate $C_2$, and the null hypothesis is $H_0: \beta_1 = 0$. The permutation of residuals under the null model approach would permute residuals from the "null" model, $Y = \beta_0 + \beta_2 C_2$, where $\beta_1$ is assumed to be 0. The permutation of residuals under the full model approach would permute residuals from the "full" model $Y = \beta_0 + \beta_1 C_1 + \beta_2 C_2$, under the alternative assumption that $\beta_1 \neq 0$.

Existing methods for permutation testing in mediation analysis for studies with small sample sizes permute the residuals under the full model or the raw data (*Koopman et al., 2015*; *Taylor & MacKinnon, 2012*). In order to accommodate covariates, we propose two additional permutation approaches for testing the indirect effect in mediation analysis based on permutation of residuals under the reduced model: (1) the permutation test of the Indirect Effect under Reduced Models (IERM), (2) the Permutation Supremum test
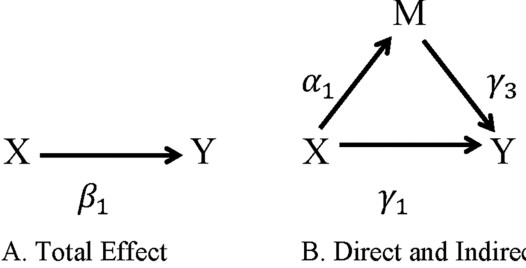

**Figure 1 Direct and indirect effects of X on Y through M.** (A) The total effect of exposure $X$ on outcome $Y$. (B) The indirect effect of the exposure $X$ on the outcome $Y$ through the mediator $M$ is represented by the path $X \to M \to Y$ and the direct effect of the exposure $X$ on the outcome $Y$ not through the mediator $M$ is represented by the path $X \to Y$.

under Reduced Models (PSRM). Through simulation studies, we compare these two proposed approaches to the standard permutation approach for mediation analysis in studies with small sample sizes, the permutation test of the Indirect Effect under Full Models (IEFM) (*Koopman et al., 2015*; *Taylor & MacKinnon, 2012*). We consider test performance in the presence of covariates since mediation analysis assumes no unmeasured confounders of the exposure–mediator–outcome relationship. We illustrate the potential consequences of permuting without careful consideration of the underlying assumptions and the null hypothesis, and provide recommendations on using permutation testing for mediation analysis in studies with small sample sizes.

## METHODS

Regression-based approaches for evaluating mediation were first popularized by *Baron & Kenny (1986)*, and extensions are now widely used in psychology and epidemiology research (*MacKinnon & Fairchild, 2009*; *Vanderweele & Vansteelandt, 2009*). Let $X$ be the exposure or intervention variable, $M$ the mediating or intermediate variable, $Y$ the outcome of interest, and $C$ a set of covariates. The relationships between these variables are illustrated in Fig. 1 and defined in the following equations for the outcome $Y$ (Eq. 1) and the mediator $M$ (Eq. 2).

$$E[Y|X = x, \ C = c, \ M = m] = \gamma_0 + \gamma_1 x + \gamma_2 c + \gamma_3 m \tag{1}$$

$$E[M|X = x, \ C = c] = \alpha_0 + \alpha_1 x + \alpha_2 c \tag{2}$$

where the indirect effect (IE) can be calculated such that

$$\widehat{IE} = \hat{\alpha}_1 * \hat{\gamma}_3 \tag{3}$$

We propose two permutation approaches to test the indirect effect, which permutes the residuals under the reduced model (*Freedman & Lane, 1983*) and have not been previously implemented for mediation analysis. We compare these two proposed methods to the standard method for permutation testing in mediation analysis for studies with small sample sizes, which permutes the residuals under the full model (*Koopman et al., 2015*; *Taylor & MacKinnon, 2012*). The methods for all three approaches can be used for both

exact and approximate permutation tests, depending on the number of permutations specified ($n_{\text{perm}}$). An exact permutation test calculates a test statistic for all possible permuted samples, whereas an approximate permutation test calculates a subset of samples.

## Proposed approach 1: permutation test of the IERM

The permutation test of the IERM makes use of the Freedman and Lane approach for permuting the residuals (*Freedman & Lane, 1983*) from the reduced model. Using their general framework, the IERM approach fits full models for Eqs. (1) and (2) in order to obtain an estimate for the indirect effect (Eq. 3). Two reduced models are fit excluding the parameters used to calculate the indirect effect and residuals from each are calculated and permuted. This approach is intended to break the associations between $M$ and $Y$, and $X$ and $M$, without disturbing associations with $C$, thus creating an estimate of the null distributions for $\alpha_1$ and $\gamma_3$, and of $\alpha_1 * \gamma_3$.

The proposed IERM approach implements the following steps:

1. Fit the full models: $Y = \hat{\gamma}_0 + \hat{\gamma}_1 X + \hat{\gamma}_2 C + \hat{\gamma}_3 M + e_Y$ and
   $M = \hat{\alpha}_0 + \hat{\alpha}_1 X + \hat{\alpha}_2 C + e_M$
2. Estimate the original estimate for the indirect effect, $(\hat{\alpha}_1 * \hat{\gamma}_3)_{\text{orig}}$
3. Fit reduced models: $Y = \hat{\gamma}_{0(r)} + \hat{\gamma}_{1(r)} X + \hat{\gamma}_{2(r)} C + e_{Y(r)}$ and
   $M = \hat{\alpha}_{0(r)} + \hat{\alpha}_{2(r)} C + e_{M(r)}$
4. Using the reduced models from Step 3, estimate $\hat{Y}$, $e_{Y(r)}$, $\hat{M}$, and $e_{M(r)}$
5. Permute residuals from the reduced models, now labeled $e_Y^*$ and $e_M^*$, $n_{\text{perm}}$ times and for each permutation, calculate $Y^* = \hat{Y} + e_Y^*$ and $M^* = \hat{M} + e_M^*$.
6. For each permutation, fit the regression models from step 1, replacing $Y$ and $M$ with $Y^*$ and $M^*$ respectively. Calculate $\hat{\alpha}_1^* * \hat{\gamma}_3^*$.
7. The absolute value of $(\hat{\alpha}_1 * \hat{\gamma}_3)_{\text{orig}}$ is compared to the distribution of absolute values of $\hat{\alpha}_1^* * \hat{\gamma}_3^*$ (for a two-tailed test). The $p$-value is calculated as the proportion of values in the distribution that have equal or greater magnitudes than $(\hat{\alpha}_1 * \hat{\gamma}_3)_{\text{orig}}$, that is, whose absolute values are greater than or equal to the absolute value of $(\hat{\alpha}_1 * \hat{\gamma}_3)_{\text{orig}}$.

## Proposed approach 2: permutation supremum test under reduced models

The PSRM also makes use of the Freedman and Lane permutation approach (*Freedman & Lane, 1983*), and incorporates a supremum testing approach for mediation by *Wagner et al. (2017)* to evaluate the composite null hypothesis $\alpha_1 * \gamma_3 = 0$, $\alpha_1 = 0$ and $\gamma_3 = 0$. The PSRM differs from the IERM above in that the individual coefficients along the proposed pathway, $\hat{\alpha}_1$ and $\hat{\gamma}_3$, are tested along with the indirect effect. In order to conclude that mediation is present, $\hat{\alpha}_1 * \hat{\gamma}_3$, $\hat{\alpha}_1$ and $\hat{\gamma}_3$ must be significantly different from zero.

The proposed PSRM differs from the IERM in Steps 2, 6 and 7, and implements the following steps:

1. Fit the full models: $Y = \hat{\gamma}_0 + \hat{\gamma}_1 X + \hat{\gamma}_2 C + \hat{\gamma}_3 M + e_Y$ and
   $M = \hat{\alpha}_0 + \hat{\alpha}_1 X + \hat{\alpha}_2 C + e_M$

2. Estimate the original estimates $\hat{\alpha}_{1\text{orig}}$, $\hat{\gamma}_{3\text{orig}}$ and $(\hat{\alpha}_1 * \hat{\gamma}_3)_{\text{orig}}$.

3. Fit reduced models: $Y = \hat{\gamma}_{0(r)} + \hat{\gamma}_{1(r)}X + \hat{\gamma}_{2(r)}C + e_{Y(r)}$ and
$M = \hat{\alpha}_{0(r)} + \hat{\alpha}_{2(r)}C + e_{M(r)}$

4. Using the reduced models from Step 3, estimate $\hat{Y}$, $e_{Y(r)}$, $\hat{M}$, and $e_{M(r)}$

5. Permute residuals from the reduced models, now labeled $e_Y^*$ and $e_M^*$, $n_{\text{perm}}$ times and for each permutation, calculate $Y^* = \hat{Y} + e_Y^*$ and $M^* = \hat{M} + e_M^*$.

6. For each permutation, fit the regression models from step 1, replacing $Y$ and $M$ with $Y^*$ and $M^*$ respectively. Estimate $\hat{\alpha}_1^* * \hat{\gamma}_3^*$, $\hat{\alpha}_1^*$ and $\hat{\gamma}_3^*$.

7. The absolute value of $(\hat{\alpha}_1 * \hat{\gamma}_3)_{\text{orig}}$ is compared to the distribution of absolute values of $\hat{\alpha}_1^* * \hat{\gamma}_3^*$ (for a two-tailed test). The $p$-value is calculated as the proportion of values in the distribution that have equal or greater magnitudes than $(\hat{\alpha}_1 * \hat{\gamma}_3)_{\text{orig}}$. The absolute value of $\hat{\alpha}_{1\text{orig}}$ is compared to the distribution of absolute values of $\hat{\alpha}_1^*$ (for a two-tailed test). The $p$-value is calculated as the proportion of values in the distribution that have equal or greater magnitudes than $\hat{\alpha}_{1\text{orig}}$. Similarly, a $p$-value is obtained for $\hat{\gamma}_{3\text{orig}}$. The null hypothesis is rejected only if $(\hat{\alpha}_1 * \hat{\gamma}_3)_{\text{orig}}$, $\hat{\alpha}_{1\text{orig}}$ and $\hat{\gamma}_{3\text{orig}}$ are significantly different from zero, and the significance level is the supremum of the significance levels of the individual tests.

## Approach for comparison: the permutation test of the IEFM

The permutation test for the IEFM makes use of ter Braak's method (*Manly, 1997*; *ter Braak, 1992*), fitting full models for both regression models and estimating the indirect effect. Rather than estimating both full and reduced models as in the PSRM and IEFM, only the full models are fitted in this approach. Residuals are permuted and used to create a sampling distribution of the test statistic and estimate confidence limits for the indirect effect (*Koopman et al., 2015*; *Taylor & MacKinnon, 2012*).

The IEFM method implements the following steps:

1. Fit the full models: $Y = \hat{\gamma}_0 + \hat{\gamma}_1 X + \hat{\gamma}_2 C + \hat{\gamma}_3 M + e_Y$ and
$M = \hat{\alpha}_0 + \hat{\alpha}_1 X + \hat{\alpha}_2 C + e_M$

2. Estimate the original estimate for the indirect effect $(\hat{\alpha}_1 * \hat{\gamma}_3)_{\text{orig}}$

3. Permute residuals from the full models, now labeled $e_Y^*$ and $e_M^*$, $n_{\text{perm}}$ times and for each permutation, calculate $Y^* = \hat{Y} + e_Y^*$ and $M^* = \hat{M} + e_M^*$.

4. For each permutation, fit the regression models from step 1, replacing $Y$ and $M$ with $Y^*$ and $M^*$ respectively. Calculate $\hat{\alpha}_1^* * \hat{\gamma}_3^*$.

5. Confidence bounds are estimated using the $\left(\frac{\omega}{2}\right) * 100$ and $\left(1 - \frac{\omega}{2}\right) * 100$ percentiles of the distribution of $\hat{\alpha}_1^* * \hat{\gamma}_3^*$, where $\omega$ corresponds to the desired alpha level.

6. The null hypothesis $(\hat{\alpha}_1 * \hat{\gamma}_3)_{\text{orig}} = 0$, is rejected if 0 is not contained within the confidence bounds.

## Simulation studies

Simulation studies were used to empirically evaluate the behavior of the two proposed approaches (IERM and PSRM) and the existing IEFM method (*Koopman et al., 2015*;

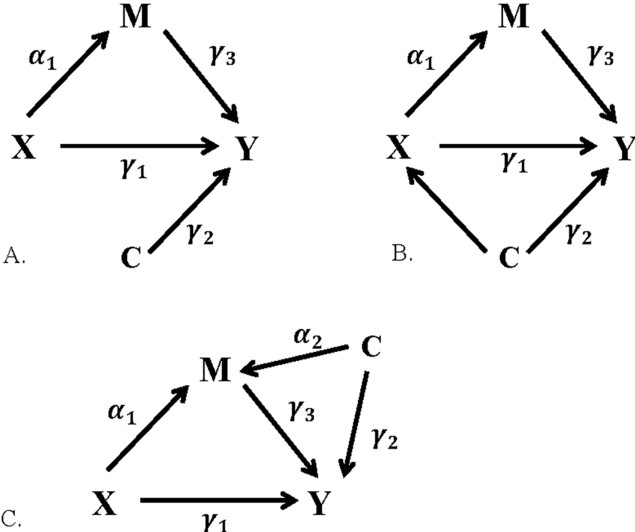

**Figure 2 Relationships between exposure, mediator, outcome, and covariate.** (A) *C* as a covariate, (B) *C* as a confounder of the *X–Y* relationship, (C) *C* as a confounder of the *M–Y* relationship.

*Taylor & MacKinnon, 2012*). To ensure scenarios of weak and strong confounding, normally distributed data were generated for *X*, *M*, *Y* and *C* from a range of correlation structures by multiplying the Cholesky decomposed matrix of the correlation structure with a matrix of independent and random normally distributed values. A subset of all possible correlations of 0, 0.15, 0.3 and 0.6 between the four variables with positive definite correlation structures was evaluated (see Tables S1 and S2 for complete list of scenarios and the corresponding regression coefficients). Correlation strengths were selected to simulate comparable regression coefficients to prior studies on permutation methods for mediation (*Taylor & MacKinnon, 2012*); the simulated regression coefficients are reported in the Online Supplemental 1. For example, one condition evaluated was a scenario where *C* was a weak confounder between *X* and *Y*, with a correlation structure of

$$
\begin{array}{c}
X \\
C \\
M \\
Y
\end{array}
\begin{bmatrix}
1 & 0.15 & 0.15 & 0.15 \\
\ldots & 1 & 0 & 0.15 \\
\ldots & \ldots & 1 & 0.15 \\
\ldots & \ldots & \ldots & 1
\end{bmatrix}
$$

Three covariate or confounding scenarios were considered in this study: (1) *C* as a covariate, associated with outcome *Y* but not with the exposure *X* or mediator *M* (Fig. 2A), (2) *C* as a confounder of the exposure–outcome (*X–Y*) relationship (Fig. 2B), and (3) *C* as a confounder of the mediator–outcome (*M–Y*) relationship (Fig. 2C). For each scenario, *C* was assessed as either a "weak" or "strong" covariate or confounder, with correlation of 0.15 or 0.6 respectively, resulting in a total of six covariate scenarios evaluated.

Under the null of no indirect effect, type I error rates were evaluated at a significance level of $\alpha = 0.05$. The null hypothesis of no indirect effect may be simulated from the following three scenarios: (1) both $\alpha_1 = 0$ and $\gamma_3 = 0$, (2) $\alpha_1 = 0$ and $\gamma_3 \neq 0$, or (3) $\alpha_1 \neq 0$ and $\gamma_3 = 0$. All three null possibilities were evaluated for *C* as a weak or strong

covariate or confounder. All other correlations were held at 0.15 for the "weak" scenario, and 0.6 for the "strong" scenario, resulting in a total of 27 conditions (13 conditions for scenario $a$, eight conditions for scenario $b$, and six conditions for scenario $c$) per sample size being evaluated under the null.

Under the alternative hypothesis, both $\alpha_1 \neq 0$ and $\gamma_3 \neq 0$. Correlation between the exposure $X$ and the mediator $M$, $\rho_{XM}$, or correlation between the mediator $M$ and outcome $Y$, $\rho_{MY}$, varied from 0.15, 0.3 and 0.6. For each of the six covariate scenarios, $\rho_{XM}$ and $\rho_{MY}$ were varied as 0.15, 0.3 or 0.6, while all other correlations were held at 0.15 for the "weak" scenario, and 0.6 for the "strong" scenario, resulting in a total of 51 conditions (18 conditions for scenario $a$, 18 conditions for scenario $b$, and 15 conditions for scenario $c$) per sample size being evaluated under the alternative. Similar to *Koopman et al. (2015)*, we considered sample sizes of 30 and 100. For each simulation condition, 1,000 replicates were run with 10,000 permutations per replicate. Simulation code is available at https://github.com/kroehlm/Permutation_Mediation_Test.

## RESULTS

Across the three covariate scenarios, results were very similar for conditions in which the coefficients $\alpha_1$ and $\gamma_3$ were mirrored; that is, the results when $\alpha_1 = 0.15$ and $\gamma_3 = 0$ were similar to those for $\alpha_1 = 0$ and $\gamma_3 = 0.15$. For simplicity, only the results for one set of covariate scenarios, $C$ as a confounder of the exposure–outcome ($X$–$Y$) relationship, will be presented; results from all scenarios are in the Online Supplemental 1.

### Type I error

Type I error rates with $C$ as a confounder between the exposure–outcome ($X$–$Y$) relationship are shown in Table 1. Results are shown in Table 1 for the null scenario (1) in which no relationship exists between the exposure–mediator ($X$–$M$) and mediator-outcome ($M$–$Y$) (i.e., $\alpha_1 = 0$, $\gamma_3 = 0$) and the null scenario (2) where no relationship exists between the exposure–mediator ($X$–$M$) (i.e., $\alpha_1 = 0$ and $\gamma_3 \neq 0$). The null scenario (3) where no relationship exists between mediator-outcome ($M$–$Y$) (i.e., $\alpha_1 \neq 0$ and $\gamma_3 = 0$) are available in the Online Supplemental 1. Each row of Table 1 represents a single simulation condition, and for each condition, the table includes the correlation conditions among all variables, the corresponding $\alpha_1$ and $\gamma_3$ coefficient values, and the empirical type I error rates for the three different permutation applications.

Type I rates were generally slightly higher when $C$ was a strong confounder compared to when it was a weak confounder. The results in the first row of Table 1 represent the scenario where $C$ is a weak confounder between the $X$ and $Y$ relationship, the type I error rate for the test of the indirect effect under the reduced models, the IERM approach, was 0.044. The proposed PSRM approach and the existing IEFM approach had more conservative type I error rates of 0.005 and 0.006, respectively. Rates for all three methods increased for conditions with a strong confounder.

The IERM approach had type I rates at the nominal level when both $\alpha_1$ and $\gamma_3$ were zero. When either $\alpha_1$ or $\gamma_3$ was nonzero, the type I error rates exceeded the nominal levels, with rates increasing with increasing size of the nonzero coefficient, and reaching

**Table 1 Type I error rates for $C$ as a confounder between the $X$–$Y$ relationship.** Under 1,000 simulations with a binary outcome of accept or reject, deviations in type I error rates beyond a Wald confidence interval (0.036, 0.064) suggest deviations from a level 0.05 test not due to sampling. Scenarios where the error rates exceeded the Wald confidence bounds are bolded.

| | Conditions | | | | | | | | Type I error rates | | |
|---|---|---|---|---|---|---|---|---|---|---|---|
| | $\rho_{XC}$ | $\rho_{XM}$ | $\rho_{CM}$ | $\rho_{XY}$ | $\rho_{CY}$ | $\rho_{MY}$ | $\alpha_1$ | $\gamma_3$ | IERM | PSRM | IEFM |
| $n = 30$ | | | | | | | | | | | |
| Weak confounder | | | | | | | | | | | |
| $\alpha_1 = 0, \gamma_3 = 0$ | | | | | | | | | | | |
| 1. | 0.15 | 0 | 0 | 0.15 | 0.15 | 0 | 0 | 0 | 0.044 | 0.005 | 0.006 |
| $\alpha_1 = 0, \gamma_3 \neq 0$ | | | | | | | | | | | |
| 2. | 0.15 | 0 | 0 | 0.15 | 0.15 | 0.15 | 0 | 0.15 | **0.084** | 0.006 | 0.007 |
| 3. | 0.15 | 0 | 0 | 0.15 | 0.15 | 0.3 | 0 | 0.3 | **0.196** | 0.016 | 0.016 |
| 4. | 0.15 | 0 | 0 | 0.15 | 0.15 | 0.6 | 0 | 0.6 | **0.499** | 0.059 | **0.075** |
| Strong confounder | | | | | | | | | | | |
| $\alpha_1 = 0, \gamma_3 = 0$ | | | | | | | | | | | |
| 5. | 0.6 | 0 | 0 | 0.6 | 0.6 | 0 | 0 | 0 | **0.070** | 0.006 | 0.008 |
| $\alpha_1 = 0, \gamma_3 \neq 0$ | | | | | | | | | | | |
| 6. | 0.6 | 0 | 0 | 0.6 | 0.6 | 0.15 | 0 | 0.15 | **0.127** | 0.010 | 0.011 |
| 7. | 0.6 | 0 | 0 | 0.6 | 0.6 | 0.3 | 0 | 0.3 | **0.300** | 0.028 | 0.035 |
| 8. | 0.6 | 0 | 0 | 0.6 | 0.6 | 0.6 | 0 | 0.6 | **0.653** | **0.068** | **0.090** |
| $n = 100$ | | | | | | | | | | | |
| Weak c onfounder | | | | | | | | | | | |
| $\alpha_1 = 0, \gamma_3 = 0$ | | | | | | | | | | | |
| 9. | 0.15 | 0 | 0 | 0.15 | 0.15 | 0 | 0 | 0 | **0.081** | 0.002 | 0.002 |
| $\alpha_1 = 0, \gamma_3 \neq 0$ | | | | | | | | | | | |
| 10. | 0.15 | 0 | 0 | 0.15 | 0.15 | 0.15 | 0 | 0.15 | **0.177** | 0.014 | 0.010 |
| 11. | 0.15 | 0 | 0 | 0.15 | 0.15 | 0.3 | 0 | 0.3 | **0.452** | 0.044 | 0.043 |
| 12. | 0.15 | 0 | 0 | 0.15 | 0.15 | 0.6 | 0 | 0.6 | **0.719** | 0.063 | **0.068** |
| Strong confounder | | | | | | | | | | | |
| $\alpha_1 = 0, \gamma_3 = 0$ | | | | | | | | | | | |
| 13. | 0.6 | 0 | 0 | 0.6 | 0.6 | 0 | 0 | 0 | 0.049 | 0.003 | 0.003 |
| $\alpha_1 = 0, \gamma_3 \neq 0$ | | | | | | | | | | | |
| 14. | 0.6 | 0 | 0 | 0.6 | 0.6 | 0.15 | 0 | 0.15 | **0.255** | 0.026 | 0.021 |
| 15. | 0.6 | 0 | 0 | 0.6 | 0.6 | 0.3 | 0 | 0.3 | **0.583** | 0.047 | 0.048 |
| 16. | 0.6 | 0 | 0 | 0.6 | 0.6 | 0.6 | 0 | 0.6 | **0.783** | 0.051 | 0.056 |

levels as high as 0.653 for a sample size of 30 and 0.783 for a sample size of 100. Both the proposed PSRM approach and the existing IEFM approach had type I error rates well below the nominal level when both $\alpha_1$ and $\gamma_3$ were zero. When either $\alpha_1$ or $\gamma_3$ was nonzero, the type I error rates increased with increasing size of the nonzero coefficient, and was generally around the nominal level when the nonzero coefficient was large. However, when the sample size was small (i.e., 30), nominal levels were exceeded a handful of times for the IEFM approach and only once for the proposed PSRM approach. For a larger

sample size, type I error rates remained below any expected deviations from nominal levels with one exception; the error rates were slightly inflated for the scenario where $C$ was a weak confounder between $X$ and $Y$, and $\gamma_3$ was large for the IEFM test (condition 12 in Table 1) but not for the proposed PSRM approach. Across the 54 null conditions evaluated, type I error rates were exceeded in 3.7% (2) of the conditions for the PSRM, and in 14.8% (8) of the conditions for the IEFM. With a smaller sample size, the IEFM approach occasionally had slightly higher rates than those for the proposed PSRM approach. However, for the larger size of $n = 100$, the error rates between the two tests were very similar, with neither consistently being larger than the other.

## Power

Results for power when $C$ was a confounder between the $X$ and $Y$ relationship are shown in Table 2 (similar results for the other covariate scenarios are in Tables S5 and S6). Due to the highly inflated error rates for the proposed IERM approach, we excluded results of power from this approach. As with the type I error rates, there were no major differences of power based on the covariate scenario within a permutation method. As with Table 1, each row represents a single simulation condition, and for each condition, the table includes the correlations among all variables, the corresponding $\alpha_1$ and $\gamma_3$, and the power for the two permutation approaches. The results in the first row represent the scenario where $C$ is a weak confounder between the $X$ and $Y$ relationship, under the alternative condition that both $\alpha_1$ and $\gamma_3$ are nonzero. As a weak confounder between $X$ and $Y$, the correlations between $X$–$C$ and $C$–$Y$ were simulated to be 0.15, with zero correlation between $C$ and $M$. The correlation between $X$ and $Y$ was also simulated to be 0.15. The correlations between $X$–$M$ and $M$–$Y$ were both simulated to be 0.15, with corresponding coefficients of $\alpha_1 = 0.1535$ and $\gamma_3 = 0.1335$. For this condition, the power for the proposed PSRM approach was 0.014, and power for IEFM approach was 0.019.

As expected, power was lowest for small values of a coefficient, and increased with increasing values of the coefficients. For the small sample size (i.e., $n = 30$), power was typically larger for the IEFM approach, especially for larger coefficient values. In the 51 alternative conditions evaluated for this sample size, power for the PSRM was only equal to the IEFM in one condition, and was, on average, 2.2% lower than power for the IEFM. Once the sample size was increased to 100, however, there were no differences in power between the two tests; see Fig. 3. Power for the PSRM was equal or better than the IEFM in 25 of the 51 conditions for this sample size, with a mean difference between the two methods of 0.31%. Figure 4 displays the change in power of the proposed PSRM approach for both weak and strong confounders, based on their coefficient values, for $n = 100$ where $C$ was a confounder between the $X$ and $Y$ relationship.

## Comparison to bootstrap based approaches for mediation analysis

Prior evaluations of the type I error rates for the IEFM approach were not inflated as indicated by simulation studies (*Koopman et al., 2015*; *Taylor & MacKinnon, 2012*). Upon completion of our studies and finding the IEFM and PSRM did exceed nominal rates under a small set of conditions, we conducted a post-hoc study to compare these methods

**Table 2 Power for C as a confounder between the X–Y relationship.**

| Conditions | | | | | | | | Power | |
| --- | --- | --- | --- | --- | --- | --- | --- | --- | --- |
| $\rho_{XC}$ | $\rho_{XM}$ | $\rho_{CM}$ | $\rho_{XY}$ | $\rho_{CY}$ | $\rho_{MY}$ | $\alpha_1$ | $\gamma_3$ | PSRM | IEFM |
| *n* = 30 | | | | | | | | | |
| Weak confounder | | | | | | | | | |
| 0.15 | 0.15 | 0 | 0.15 | 0.15 | 0.15 | 0.1535 | 0.1335 | 0.014 | 0.019 |
| 0.15 | 0.15 | 0 | 0.15 | 0.15 | 0.3 | 0.1535 | 0.287 | 0.056 | 0.062 |
| 0.15 | 0.15 | 0 | 0.15 | 0.15 | 0.6 | 0.1535 | 0.5941 | 0.119 | 0.149 |
| 0.15 | 0.3 | 0 | 0.15 | 0.15 | 0.15 | 0.3069 | 0.1221 | 0.043 | 0.048 |
| 0.15 | 0.3 | 0 | 0.15 | 0.15 | 0.3 | 0.3069 | 0.2873 | 0.125 | 0.134 |
| 0.15 | 0.3 | 0 | 0.15 | 0.15 | 0.6 | 0.3069 | 0.6177 | 0.363 | 0.413 |
| 0.15 | 0.6 | 0 | 0.15 | 0.15 | 0.15 | 0.6138 | 0.1136 | 0.074 | 0.104 |
| 0.15 | 0.6 | 0 | 0.15 | 0.15 | 0.3 | 0.6138 | 0.351 | 0.313 | 0.354 |
| 0.15 | 0.6 | 0 | 0.15 | 0.15 | 0.6 | 0.6138 | 0.8259 | 0.931 | 0.951 |
| Strong confounder | | | | | | | | | |
| 0.6 | 0.15 | 0 | 0.6 | 0.6 | 0.15 | 0.2344 | 0.0972 | 0.016 | 0.020 |
| 0.6 | 0.15 | 0 | 0.6 | 0.6 | 0.3 | 0.2344 | 0.2526 | 0.064 | 0.073 |
| 0.6 | 0.15 | 0 | 0.6 | 0.6 | 0.6 | 0.2344 | 0.5636 | 0.193 | 0.232 |
| 0.6 | 0.3 | 0 | 0.6 | 0.6 | 0.15 | 0.4688 | 0.0436 | 0.045 | 0.047 |
| 0.6 | 0.3 | 0 | 0.6 | 0.6 | 0.3 | 0.4688 | 0.2182 | 0.175 | 0.196 |
| 0.6 | 0.3 | 0 | 0.6 | 0.6 | 0.6 | 0.4688 | 0.5673 | 0.553 | 0.607 |
| 0.6 | 0.6 | 0 | 0.6 | 0.6 | 0.15 | 0.9375 | −0.171 | 0.127 | 0.153 |
| 0.6 | 0.6 | 0 | 0.6 | 0.6 | 0.3 | 0.9375 | 0.1714 | 0.146 | 0.181 |
| 0.6 | 0.6 | 0 | 0.6 | 0.6 | 0.6 | 0.9375 | 0.8571 | 1.000 | 1.000 |
| *n* = 100 | | | | | | | | | |
| Weak confounder | | | | | | | | | |
| 0.15 | 0.15 | 0 | 0.15 | 0.15 | 0.15 | 0.1535 | 0.1335 | 0.112 | 0.103 |
| 0.15 | 0.15 | 0 | 0.15 | 0.15 | 0.3 | 0.1535 | 0.287 | 0.263 | 0.259 |
| 0.15 | 0.15 | 0 | 0.15 | 0.15 | 0.6 | 0.1535 | 0.5941 | 0.327 | 0.340 |
| 0.15 | 0.3 | 0 | 0.15 | 0.15 | 0.15 | 0.3069 | 0.1221 | 0.159 | 0.151 |
| 0.15 | 0.3 | 0 | 0.15 | 0.15 | 0.3 | 0.3069 | 0.2873 | 0.715 | 0.707 |
| 0.15 | 0.3 | 0 | 0.15 | 0.15 | 0.6 | 0.3069 | 0.6177 | 0.888 | 0.892 |
| 0.15 | 0.6 | 0 | 0.15 | 0.15 | 0.15 | 0.6138 | 0.1136 | 0.146 | 0.160 |
| 0.15 | 0.6 | 0 | 0.15 | 0.15 | 0.3 | 0.6138 | 0.351 | 0.794 | 0.806 |
| 0.15 | 0.6 | 0 | 0.15 | 0.15 | 0.6 | 0.6138 | 0.8259 | 1.000 | 1.000 |
| Strong confounder | | | | | | | | | |
| 0.6 | 0.15 | 0 | 0.6 | 0.6 | 0.15 | 0.2344 | 0.0972 | 0.126 | 0.105 |
| 0.6 | 0.15 | 0 | 0.6 | 0.6 | 0.3 | 0.2344 | 0.2526 | 0.439 | 0.439 |
| 0.6 | 0.15 | 0 | 0.6 | 0.6 | 0.6 | 0.2344 | 0.5636 | 0.476 | 0.490 |
| 0.6 | 0.3 | 0 | 0.6 | 0.6 | 0.15 | 0.4688 | 0.0436 | 0.099 | 0.099 |
| 0.6 | 0.3 | 0 | 0.6 | 0.6 | 0.3 | 0.4688 | 0.2182 | 0.758 | 0.762 |
| 0.6 | 0.3 | 0 | 0.6 | 0.6 | 0.6 | 0.4688 | 0.5673 | 0.970 | 0.974 |
| 0.6 | 0.6 | 0 | 0.6 | 0.6 | 0.15 | 0.9375 | −0.171 | 0.323 | 0.339 |
| 0.6 | 0.6 | 0 | 0.6 | 0.6 | 0.3 | 0.9375 | 0.1714 | 0.314 | 0.326 |
| 0.6 | 0.6 | 0 | 0.6 | 0.6 | 0.6 | 0.9375 | 0.8571 | 1.000 | 1.000 |

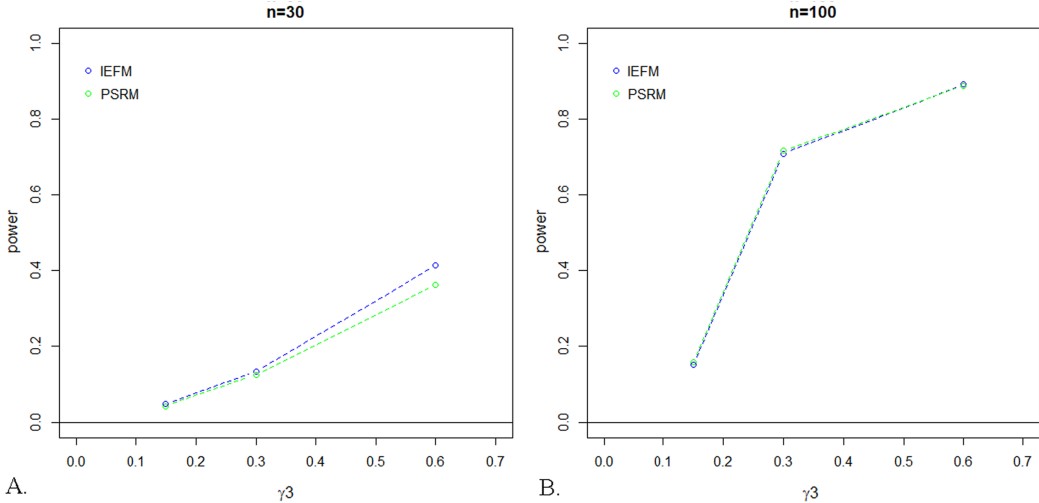

**Figure 3 Power for test of mediation given the scenario where *C* as a weak *X–Y* Confounder.** The IEFM approach is represented by the blue line and the proposed PSRM approach is represented by the green line. As seen in the plots below, the power is comparable for these two approaches for sample sizes of (A) *n* = 30 and (B) *n* = 100.

with bootstrap methods under the four most extreme null conditions we observed, as well as six alternative conditions. The purpose of these studies was to provide a side-by-side comparison of the methods reported by *Koopman et al. (2015)* and the permutation methods evaluated here under identical conditions with weak and strong confounding. Simulations were carried out as described above, and the indirect effect was evaluated by the PSRM, IEFM, and the three bootstrap methods evaluated in *Koopman et al. (2015)*: the percentile bootstrap (PB), the bias corrected bootstrap (BCB), and the bias corrected accelerated bootstrap (BCAB). With respect to type I error, results from these studies (Table 3) indicate the PB approach performs best among the bootstrap methods, while the BCB and BCAB approaches exceeded nominal levels under all four scenarios. Overall, the PSRM approach performed better than the IEFM and PB approaches, maintaining error rates at the nominal level in all but one condition. Comparisons of power among the five methods are presented in Table 4. For all conditions evaluated, power was highest for the BCB and BCAB, the two methods which exhibited inflated type I errors under all four null conditions. Power was slightly higher for the IEFM when compared with the PSRM and PB when *n* = 30, and comparable among the three methods when *n* = 100.

## Example: how framing of media stories influences attitudes regarding immigration policy

We now illustrate the application of the PSRM and IEFM with an empirical example based on the framing data of *Brader, Valentino & Suhay (2008)*. In this study, participants were randomly assigned to different media stories about immigration with either positive or negative framing and asked about their attitudes and political behavior with respect to immigration policy. They hypothesized that anxiety mediates the relationship between framing and whether a participant would agree to send a letter about immigration policy to

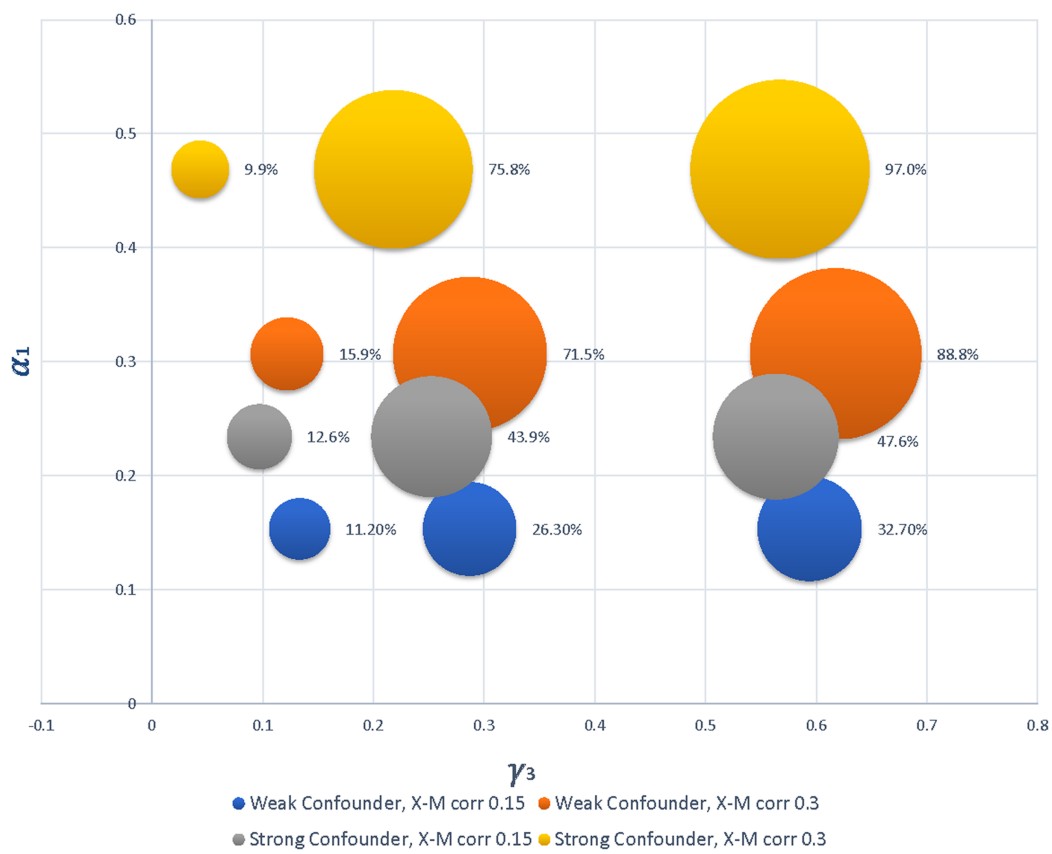

**Figure 4 Bubble plot displaying the increase in power for PSRM as strength of coefficients increase.**
The different patterns correspond to differences in the strength of the confounder and magnitude of the
correlation between *X* and *M*, and bubble size represents power. The bubble in the lowest left corner
corresponds to the first row of results for *n* = 100 in Table 2, representing *C* as a weak confounder with
correlations of *X–M* and *M–Y* simulated to be 0.15 and a power of 11.2%. Note that, because data were
generated based on correlation structures, the simulated coefficient values were sometimes different for a
weak versus strong confounder.                                     

**Table 3 Type I error rates for bootstrap simulations.** Under 1,000 simulations with a binary outcome
of accept or reject, deviations in type I error rates beyond a Wald confidence interval [0.036, 0.064]
suggest deviations from a level 0.05 test not due to sampling. Scenarios where the error rates exceeded the
Wald confidence bounds are bolded.

| Conditions | | | | | | | | Type I error rates | | | | |
|---|---|---|---|---|---|---|---|---|---|---|---|---|
| $\rho_{XC}$ | $\rho_{XM}$ | $\rho_{CM}$ | $\rho_{XY}$ | $\rho_{CY}$ | $\rho_{MY}$ | $\alpha_1$ | $\gamma_3$ | PSRM | IEFM | PB | BCB | BCAB |
| *n* = 30 | | | | | | | | | | | | |
| 0.15 | 0 | 0 | 0.15 | 0.15 | 0.6 | 0 | 0.6 | **0.071** | **0.086** | 0.065 | **0.115** | **0.109** |
| 0.6 | 0 | 0 | 0.6 | 0.6 | 0.6 | 0 | 0.6 | 0.061 | **0.079** | **0.064** | **0.100** | **0.084** |
| *n* = 100 | | | | | | | | | | | | |
| 0.15 | 0 | 0 | 0.15 | 0.15 | 0.6 | 0 | 0.6 | 0.052 | 0.056 | 0.060 | **0.090** | **0.075** |
| 0.6 | 0 | 0 | 0.6 | 0.6 | 0.6 | 0 | 0.6 | 0.051 | 0.059 | 0.060 | **0.082** | **0.065** |

**Table 4 Power for bootstrap simulations.**

| Conditions | | | | | | | | Power | | | | |
|---|---|---|---|---|---|---|---|---|---|---|---|---|
| $\rho_{XC}$ | $\rho_{XM}$ | $\rho_{CM}$ | $\rho_{XY}$ | $\rho_{CY}$ | $\rho_{MY}$ | $\alpha_1$ | $\gamma_3$ | PSRM | IEFM | PB | BCB | BCAB |
| $n = 30$ | | | | | | | | | | | | |
| 0.15 | 0.3 | 0 | 0.15 | 0.15 | 0.6 | 0.3069 | 0.6177 | 0.335 | 0.387 | 0.339 | 0.601 | 0.423 |
| 0.6 | 0.3 | 0 | 0.6 | 0.6 | 0.6 | 0.4688 | 0.5673 | 0.540 | 0.586 | 0.532 | 0.771 | 0.602 |
| 0.15 | 0.6 | 0 | 0.15 | 0.15 | 0.6 | 0.6138 | 0.8259 | 0.942 | 0.953 | 0.931 | 0.982 | 0.952 |
| $n = 100$ | | | | | | | | | | | | |
| 0.15 | 0.3 | 0 | 0.15 | 0.15 | 0.3 | 0.3069 | 0.2873 | 0.682 | 0.680 | 0.660 | 0.883 | 0.760 |
| 0.6 | 0.3 | 0 | 0.6 | 0.6 | 0.3 | 0.4688 | 0.2182 | 0.774 | 0.773 | 0.773 | 0.923 | 0.827 |
| 0.15 | 0.6 | 0 | 0.15 | 0.15 | 0.3 | 0.6138 | 0.3510 | 0.812 | 0.825 | 0.812 | 0.917 | 0.826 |

**Table 5 Results from framing of media stories analysis.**

| Education level | $n$ | IE Estimate | PSRM p-value | IEFM 95% CI |
|---|---|---|---|---|
| Less than High School | 20 | 0.343 | 0.2617 | [−0.225, 1.06] |
| High School | 92 | 0.138 | 0.0616 | [−0.006, 0.321] |
| Some College | 70 | 0.439 | 0.0030 | [0.158, 0.776] |
| Bachelor's (or higher) | 83 | 0.190 | 0.1855 | [−0.080, 0.501] |

his or her member of Congress. Their analysis, controlling for education, age, income, and sex, suggested anxiety does act as a significant mediator between the two variables. Here, we will extend the primary analysis by considering the role of framing on attitudes toward increased immigration, a four-point item with larger vales indicating more negative attitudes. Despite randomization, there was some unevenness of distribution across education groups with respect to treatment exposure, so we stratify analyses by education level. Similar to *Brader, Valentino & Suhay (2008)* we adjusted for age, income and sex.

Results from our extended analysis suggest that anxiety may mediate the relationship between framing and attitudes toward immigration in some, but not all, education groups. For subjects with a high school education, both the *p*-value from the PSRM and 95% CI from the IEFM were close to their respective decision points, but did not exceed thresholds to reject the null hypothesis (Table 5). For all education levels, inference between the two permutation methods did not differ.

## DISCUSSION

We have proposed two alternative methods for testing the presence of an indirect effect using permutation approaches under the reduced model (IERM and PSRM), and compared these approaches to an existing permutation approach for mediation analysis under the full model (IEFM) (*Koopman et al., 2015*; *Taylor & MacKinnon, 2012*). An important assumption in mediation analysis is that of no unmeasured confounders, the two methods we evaluated were based on Freedman and Lane's permutation of residuals in order to appropriately accommodate covariate adjustment. These methods have not previously been evaluated. Furthermore, our simulation studies considered

different covariate and confounding scenarios. We evaluated the different approaches to testing the indirect effect in mediation analysis with small sample sizes. While the PSRM approach has similar power to the existing IEFM approach, the proposed PSRM approach had a lower type I error rate than both the existing IEFM approach and the proposed IERM approach, especially for a small sample size of 30.

In contrast to the proposed IEFM approach, the proposed PSRM approach has the advantage of being able to directly evaluate all three components to the complex, composite null hypothesis that one or both coefficients along the mediated path are zero in order to test the presence of the indirect effect. While the proposed PSRM approach has similar or slightly less power than the existing IEFM approach (*Koopman et al., 2015*) for a sample size of 30, this was not unexpected considering the type I errors were also higher in the IEFM approach. Furthermore, for a sample size of 100, no major differences in power were noted for the proposed PSRM and the existing IEFM approaches. Further, when compared to bootstrap methods, the PSRM outperformed all three methods with respect to maintaining nominal type I error rates, and no major differences in power were found between the PSRM and the PB, which was the only bootstrap method that did not exceed nominal type I error rates under all conditions evaluated.

The proposed IERM approach is not recommended due to the inflated type I error rate. The poor performance of the IERM approach is due to discrepancy between the composite hypothesis of the indirect effect and the null hypothesis under which permutation was performed. The null hypothesis for the evaluation of the indirect effect is written as: $Ho: \alpha_1 * \gamma_3 = 0$. However, by using the permutation under reduced models approach, the $X$–$M$ and $M$–$Y$ associations were broken up in their respective regression models, and permutation was performed under the null hypothesis that $\alpha_1 = 0$ and $\gamma_3 = 0$. Thus, the $p$-value is the probability of exceeding the observed value given that $\alpha_1 = \gamma_3 = 0$, not $\alpha_1 * \gamma_3 = 0$. Therefore, while the test performed well under the null hypothesis of $\alpha_1 = 0$ and $\gamma_3 = 0$, when either $\alpha_1 = 0$ and $\gamma_3 \neq 0$, or $\alpha_1 \neq 0$ and $\gamma_3 = 0$ was true, false significance for the indirect effect was achieved far too often. As noted by others, it is imperative to carefully consider and permute under the *correct* null hypothesis (*Anderson & Legendre, 1999*; *Churchill & Doerge, 2008*; *Westfall & Young, 1993*) and here we demonstrate how one may get misleading findings from a study when using an unsuitable test.

The work in this manuscript extends upon the work of *Taylor & MacKinnon (2012)* and *Koopman et al. (2015)* by examining the performance of permutation methods under a broader set of conditions including the presence of covariates and confounders. Our findings support prior works that also demonstrated permutation approaches outperform the more commonly-used bootstrap methods in terms of excess type I error rates. This study was, however, limited in scope to conditions with continuous variables and normally distributed errors, and while permutation methods would be expected to outperform many other methods under more challenging conditions (e.g., non-normally distributed errors), this remains yet to be demonstrated. Further, there is growing enthusiasm for the use of Bayesian methods as an alternative to bootstrapping when

testing mediation in small samples (*Nuijten et al., 2015b*; *Yuan & MacKinnon, 2009*), and recent contributions in this area include an R package, BayesMed, for implementation (*Nuijten et al., 2015a*). Future work in this area should further explore both permutation and Bayesian approaches under conditions with non-normal outcomes and violations to traditional regression modeling assumptions.

## CONCLUSIONS

Permutation testing has been proposed as a solution to small sample mediation testing (*Koopman et al., 2015*). Here, we evaluate two novel permutation approaches for testing the indirect effect in mediation analysis (IERM and PSRM) and compare test performance to the standard method (IEFM). The PSRM maintains nominal type I error rates under more conditions than the IEFM or bootstrap methods, and does not substantially decrease power, even in small samples. We recommend the proposed PSRM approach over the existing IEFM approach for mediation analysis in studies with small sample sizes.

## ACKNOWLEDGEMENTS

The authors thank Brent Pedersen for use of his R code to perform Freedman and Lane permutation approach.

### Funding

This work was supported by the National Institutes of Health Grant P50MH086383, NHLBI grant K01HL125858, and the Grohne-Stepp Endowment from the University of Colorado Cancer Center. The funders had no role in study design, data collection and analysis, decision to publish, or preparation of the manuscript.

### Grant Disclosures

The following grant information was disclosed by the authors:
National Institutes of Health: P50MH086383.
NHLBI: K01HL125858.
Grohne-Stepp Endowment from the University of Colorado Cancer Center.

### Competing Interests

The authors declare that they have no competing interests.

### Author Contributions

- Miranda E. Kroehl conceived and designed the experiments, performed the experiments, analyzed the data, prepared figures and/or tables, authored or reviewed drafts of the paper, and approved the final draft.
- Sharon Lutz conceived and designed the experiments, authored or reviewed drafts of the paper, and approved the final draft.
- Brandie D. Wagner conceived and designed the experiments, prepared figures and/or tables, authored or reviewed drafts of the paper, and approved the final draft.

## Data Availability

Data is available at GitHub:

https://github.com/kroehlm/Permutation_Mediation_Test/tree/add-license-1.

## Supplemental Information

Supplemental information for this article can be found online at http://dx.doi.org/10.7717/peerj.8246#supplemental-information.

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
