# Peer review of "Permutation-based methods for mediation analysis in studies with small sample sizes"

_PeerJ, doi:10.7717/peerj.8246_

## Round 0.1 · original submission · Major Revisions

The reviewers have made generally positive comments about your manuscript and provided some constructive suggestions for you to address in a revised version. Please respond to each of their comments, and my additional comments below, with a point-by-point response, explaining in each case what was changed in the manuscript or why no changes were made in response to that particular point.

I feel that your recommendations for PSRM are somewhat premature at this stage given the limited range of simulations (other parameters could also be examined) and the lack of what I would consider to be clearly superior performance. I wonder if you have considered reporting the number/percentage of scenarios from the results Tables 1 and 2 where PSRM and IEFM are the “winner” in terms of performance to help readers digest the tables more easily; or you may wish to look at other ways of further summarising these results. Is there a reason for not including the equivalent of Table 3 for power? I’d like to see more in the discussion about what the next steps, in the authors’ views, are in evaluating and improving approaches to testing for indirect effects with small samples. What should you and/or other researchers be looking at next?

Specific comments are below. Some of these are merely stylistic suggestions for you to consider and you should feel absolutely entitled to not make changes in response to these.

Line 1: Perhaps consider hyphenating “Permutation-based”.

Line 22: “where” not “were”.

Line 33: Perhaps relationships (with “s” added) as you are looking at relationships between the three variables. Same on Line 59.

Lines 38–39: For a reader of the abstract, I’m not sure that the conclusions will follow from the results. The results in the abstract do not point to any advantages of PSRM over IEFM which would justify the recommendation in the conclusions.

Line 73: I don’t think this comma is needed.

Line 90: “psychology and epidemiologic” seems mismatched to me, perhaps “psychology and epidemiology”?

Line 96: This is a terribly pedantic point, but on the left-hand side you go in the order x, m, and c; but on the RHS, you go x, c, and m (note that changing the RHS order will also change equation (3) and Lines 117, etc. below). Also, the spacing (rather the lack of spacing after the commas) differs in equation (1) compared to equation (2).

Line 124: It might be easier for a few readers to follow if you refer to the estimates for Y and M as being from Step 3 and not Step 1 (e.g. prefix this point with “Using the models from Step 3, estimate…” or “Using the reduced models, estimate…”). Same point for Line 147. Reviewer #3 also raises this point.

Line 131: Or more simply, “…that have greater magnitudes than \alpha…” Also Lines 154–155, 157–158.

Line 141: It might help some readers if you note which steps (2, 6, and 7) differ from the IERM approach above here. A similar point applies to Line 169, where it might be clearer if you note shared steps with the other approaches. Reviewer #3 mentions this point also.

Line 198: Spurious space. Also Lines 221, 226 (twice), 228, and 229.

Line 207: Can you add a parenthetical expression showing where the 27 comes from? Can you also similarly explain the 51 conditions on Line 212?

Line 219: I don’t think this is an “example” as much as it is a definition of mirroring given that a and b are variables. It would be an example if you used actual values for a and b. Reviewer #1 also comments on this part of the text.

Line 298: Perhaps “an indirect” rather than “the indirect” here.

Line 324: Spurious italics (“is”).

Line 326: Spurious “in” or missing words?

Table 1: Row 5 has inconsistent decimal places for the error rates, as do rows 6, 7, 8, and 10.

Table 1: The note “…indicate real deviations from a level 0.05 test, not due to sampling.” is slightly too strong; perhaps “…indicate evidence of real deviations from a level 0.05 test, not likely to be due to sampling variability.” (or similar). You could also argue for an exact binomial 95% CI (0.037–0.065), not that this is important for decision making here. However, if you do continue with a Wald interval (I’m guessing), please label it as such. See also the notes to Table 3. Presumably the same approach you’ve used here for formatting, etc. could also be applied to Supplementary Tables 3 and 4?

Table 2: There are a few inconsistencies in decimal places for the power here (5, I think, excluding the 1s).

Figure 4: Could you increase the legend size so that the patterns are easier to distinguish?

Reviewer 1 ·

Basic reporting

no comment

Experimental design

Some clarifications should be given why correlations of .15 and .60 were used instead of .10 and .50, which would be in line with Cohen's cutoffs for small and large effects.

Some more details should be provided about the permuation used in the current manuscript.

Validity of the findings

No comment.

Additional comments

For small sample sizes, Bayesian Analysis may be a better altertnative than bootstrapping. This could be mentioned in the discussion.

Table 2 and 3 lack the information given in the first row of table 1.

The mansucript could beneifit from a an illustrative example.

It was not clear to me what is meant by a_1 = a and gamma_3 = b and a_1 = b and gamma_3 = a in line 219 and 220.

Thank you for the opportunity to review this mansucript.

Reviewer 2 ·

Basic reporting

The manuscript is clearly written and uses common terminology in the field of mediation analysis. The manuscript consults relevant literature for the research question. The structure of the paper is clear, and the paper is self-contained.
I have a few suggestions for improvement:

- MacKinnon & Taylor (2012) make a distinction between approximate and exact permutation tests. Given the sample sizes in the current manuscript, I assume the permutation tests are approximate, but it would be nice to mention if this is the case explicitly. Furthermore, Line 282 incorrectly states that Taylor and MacKinnon (2012) did not report Type I error rates.

-It is unclear why the PSRM uses absolute values for Step 7 instead of raw values. Can the authors explain this choice?

Experimental design

The research question is defined clearly, and there is no doubt that this paper fills a gap in the literature. The choice of the set of models and conditions to examine in the simulation study is appropriate. Parameter combinations are described in sufficient detail for other researchers to replicate the simulation study, however, it would be nice to include code for executing the analyses using each method tested in the simulation study.

Validity of the findings

The conclusions are well stated, linked to original research question and limited to supporting results.
The findings in Lines 291-293 confirm findings from previous literature about the bias-corrected bootstrap having excessive Type I error rates (MacKinnon, Lockwood, & Williams, 2004; Fritz, Taylor, & MacKinnon, 2012), and the citations of papers that found the same should be included.

Recommended reference to add:

Fritz, M. S., Taylor, A. B., & MacKinnon, D. P. (2012). Explanation of two anomalous results in statistical mediation analysis. Multivariate behavioral research, 47(1), 61-87.

Additional comments

More details about how to interpret Figure 4 (e.g., picking one bubble and describing what it’s size and location mean) would be helpful .

Reviewer 3 ·

Basic reporting

This is all fine.

Experimental design

I wonder what the usefulness of presenting the IERM is (See general comments).

Validity of the findings

Please supply computer code used to verify validity and reproducibility.

Additional comments

General Points

1. What’s the difference between IERM and PSRM other than the additional tests on alpha and gamma at the end in step 7? If that’s the only difference then
a. just say so and omit the re-explanation of the steps.
b. If the additional tests for alpha and gamma are needed to guarantee the mediation effect is zero, as stated in line 140, then is IERM even useful? Why would you ever use this test?

2. Lines 327-334: If you knew that IERM was unsuitable a priori then why propose it as a method in the first place? Your arguments in this paragraph seem be sufficient to rule it out as a useful tool for the purposes of testing your target hypothesis so why bother with the permutation tests? Why not just give readers the logical explanation and done?

3. Line 272: With power so low, are these tests any use at all?! Would you recommend the, as useful in practice?

4. Is 100 a “small” sample size in psychology experiments? What is a typical sample size in the kinds of studies this work is aimed at?

5. Why does the bootstrap lead to biased results? What is it about the permutation tests that makes them free from this particular shortcoming?

6. Could you supply computer code for your simulations? Given that the results are specific to the inputs it would be useful if readers wish to apply your tests to their own studies with different characteristics.


Specific Points

7. Line 100 (or vicinity): State the hypotheses explicitly , I.e., H0 = … and H1 = … . This will make it obvious what you mean by ‘null’ and ‘full’ model.
8. Add C to Fig 1.
9. Vanderweele and Vansteelandt, 2009 not in references.
10. Line 124: Yhat and Mhat are from the full or reduced model?
11. Line 129: What is gamma3 orig? This symbol was not introduced.
12. Line 202: “significant” should be “significance”.

---

## Round 0.2 · Minor Revisions

Thank you for your constructive revisions. Both reviewers are happy with your manuscript, with a small suggestion from Reviewer 3 about referencing your own GitHub repository, which I think readers would like to see included in your manuscript itself. I’ll also make a small number of suggestions below to tidy up a few possible wording and formatting issues that remained. If there were slightly fewer of these, I’d recommend that your manuscript be accepted now and leave these to be addressed in proofing, but there are perhaps just enough of these to correct now and then I will be delighted to recommend your manuscript be accepted.

Line 32: I’m comfortable with “unmeasured” here, but on re-reading, it seems to me that more precise would be “unmodelled”. Also Lines 60, 91, and 349. I’ll leave this point entirely to your discretion.

Line 88: This seems to be the first use of IEFM in the body of the manuscript and should be given in full for this first use. I appreciate that this expansion is also included in the abstract, but it is possible for a reader to skip/only skim the abstract. You eventually give the abbreviation in full on Lines 176 (heading) and 177 (in text), and the text on Line 177 could simply be moved to this earlier first use, but note that you say “model” there and not “models” as you do on Line 176, and add “the” (“…under the full model…”) which is not elsewhere included, i.e. Lines 30 and 175. I appreciate that this is very pedantic, but it would be good to have the same expressions used throughout.

Line 100: There appears to be a spurious closing parenthesis in “(Mackinnon & Fairchild, 2009; VanderWeele, T, 2009)).”

Lines 143–144: I think there is a minor issue here that would, for all intent and purposes, never arise in practice due to modern numerical precision, but you say “that have greater magnitudes” (strict inequality) and then “are greater than or equal to” (non-strict inequality), counting the equal case in the latter but not the former. The equal to case should be included as per the definition of a p-value, so perhaps “…that have EQUAL OR greater magnitudes…” on Line 143? The same applies to Lines 168, and 170–171.

Line 172: I think there is a missing space after “if” before the inline equation.

Line 192: Should “percentile” here be “confidence interval percentage” as you’re talking about the CI coverage? For omega=0.05, 1-0.05 would be a 95 percent CI (rather than a 95th percentile)?

Line 210: The last column of the matrix seems out of alignment for the final two row.

Line 225: The parenthesis should perhaps remain “attached” to the contents on Line 226.

Line 248: I think you want a comma between “condition” and “the” here. Also on Line 282.

Lines 261–262: There seemed to be a spurious page break here.

Line 316: I’m assuming the “BP” here should be “PB”.

Line 333: Do you mean “…a four-point ITEM with…”?

Line 364: I think you mean “approachES” here.

Line 394: Could you add a reference here (for the R package, which I assume is BayesMed, e.g. to its CRAN page)?

The figures are very slightly blocky for me (all four figures). If this is also the case for you, do you have higher resolution versions of the PNG files?

Reviewer 1 ·

Basic reporting

no comment

Experimental design

no comment

Validity of the findings

no comment

Additional comments

I was a reviewer of a previous version of the manuscript and found the authors were responsive in addressing the concerns raised in the previous round by the editor and the reviewers. The current version is an improvement over the previous one. This work will make an important contribution to the literature on mediation analysis. Thus, I strongly suggest to accept this manuscript for publication in its current form.

Reviewer 3 ·

Basic reporting

- Please reference the github repository with your code in the manuscript so future readers have access.

Experimental design

Fine

Validity of the findings

Fine

---

## Round 0.3 · accepted · Accept

Thank you for your revisions. I am very happy to accept your manuscript and look forward to seeing the methods and ideas in your article being discussed and used by researchers. I have noted a small number of minor typographical issues that I will leave for you to address as you see best when proofing your manuscript.

In the tracked changes version, you’re sometimes inconsistent with spaces between sentences, mixing one and two spaces throughout (e.g., most of the introduction uses single spaces but there are a few double spaces towards the end). There are also some spurious double spaces within list items and sentences (Lines 133, 136, 151, 161, 165, 192, 248, and 272 in this version, and perhaps elsewhere).

Line 76 (PDF version): “…the hull hypothesis…”

Line 115 (PDF version): You say “all 3 approaches” here, but used “these two proposed methods” on Line 112 (see many other such examples in the manuscript of small numbers being presented in words). I suggest spelling out numbers ten and below, but this should be consistent in any case. See also Lines 87, 216, 222, 223, 226, and 318.

Line 118 (PDF version): I think this would be clearer with “this for” added to “calculates THIS FOR a subset of samples.” (you are talking about calculating the test statistic and not calculating samples).

Line 283 (PDF version): You say “table 1” here but normally capitalise “Table” (c.f. Lines 243, 248, 253, 271, 279, and 320).

In the references (PDF Lines 448–450), should the package name be italicised? (This would be the title in BiBTeX using citation("BayesMed").)

Also in the references (PDF Line 461), the capitalisation of “Statistics and Its Interface” seems slightly odd to me.